



# An Overview of the E3SM version 2 Large Ensemble and Comparison to other E3SM and CESM Large Ensembles

John T. Fasullo[1], Jean-Christophe Golaz[2], Julie M. Caron[1], Nan Rosenbloom[1] Gerald A. Meehl[1],
Warren Strand[1], Sasha Glanville [1], Samantha Stevenson[3], Maria Molina[1,4], and Christine A. Shields[1],
Chengzhu Zhang[2], James Benedict[5], Tony Bartoletti[2]

[1]National Center for Atmospheric Research, Boulder, CO, 80301, USA
[2]Lawrence Livermore National Laboratory, Livermore, CA, 94551, USA
[3]Bren School of Environmental Science & Management, University of California, Santa Barbara, CA, 93117, USA
[4]University of Maryland, College Park, MD, 20742, USA
[5]Los Alamos National Laboratory, Los Alamos, NM, USA

*Correspondence to*: John T. Fasullo (fasullo@ucar.edu)

**Abstract.** This work assesses a recently produced 21-member climate model large ensemble (LE) based on the U.S.
Department of Energy's Energy Exascale Earth System Model (E3SM) version 2 (E3SM2). The ensemble spans the
historical era (1850 to 2014) and 21st Century (2015 to 2100), using the SSP370 pathway, allowing for an evaluation of the
model's forced response (FR). A companion 500-year preindustrial control simulation is used to initialize the ensemble and
estimate drift. Characteristics of the LE are documented and compared against other recently produced ensembles using the
E3SM version 1 (E3SM1) and Community Earth System Model (CESM) versions 1 and 2.

Simulation drift is found to be smaller, and model agreement with observations is higher, in versions 2 of E3SM and CESM
versus their version 1 counterparts. Shortcomings in E3SM2 include a lack of warming from the mid to late 20[th] Century
likely due to excessive cooling influence of anthropogenic sulfate aerosols, an issue also evident in E3SM1. Associated
impacts on the water cycle and energy budgets are also identified. Considerable model dependence in the FR associated with
both aerosol and greenhouse gas responses is documented and E3SM2's sensitivity to variable prescribed biomass burning
emissions is demonstrated.

Various E3SM2 and CESM2 model benchmarks are found to be on par with the highest performing recent generation of
climate models, establishing the E3SM2 LE as an important resource for estimating climate variability and responses, though
with various caveats as discussed herein. As an illustration of the usefulness of LEs in estimating the potential influence of
internal variability, the observed CERES-era trend in net top-of-atmosphere flux is compared to simulated trends and found
to be much larger than the FR in all LEs, with only a few members exhibiting trends as large as observed, thus motivating
further study.



**Short Summary**

Climate model large ensembles provide a unique and invaluable means for estimating the climate response to external forcing agents, thereby allowing for the isolation of internal variability. Here, an overview of the Energy Exascale Earth System Model (E3SM) version 2 large ensemble is given along with comparisons to large ensembles from E3SM version 1 and versions 1 and two of the Community Earth System Model. The manuscript provides broad and important context for users of these ensembles.

# 1 Introduction

Identifying the magnitude and spatiotemporal structure of the climate response to external forcing, the so-called forced-response (FR), is vital for anticipating and adapting to a changing climate (Deser et al. 2020, Huang and Stevenson, 2021, Xu et al. 2022). Single-model large ensembles (LE) consist of multiple simulations (typically ≥20) of past and future climate using prescribed emissions scenarios and initialized from similar, though not identical, climate states. Through ensemble-mean averaging, they are an important tool for FR estimation including its temporal evolution and inter-model contrasts. Relevance to nature is however limited by errors in both model physics and prescribed external forcings (Tebaldi et al. 2020, Fasullo et al. 2022). Understanding these inter-model differences and the uncertainties in forcings is key to gauging the likely range of potential outcomes under climate change.

The purpose of this work is to describe the recently produced E3SM2 LE that builds upon the initial set of simulations in Golaz et al. (2022) by adding 16 additional historical members to the original 5 and extending them all to 2100. Insights are gained by comparing this new LE with LEs using the Energy Exascale Earth System Model (E3SM) versions 1 (E3SM1, Stevenson et al., 2023) and the Community Earth System Model (CESM) versions 1 (CESM1, Kay et al. 2015) and 2 (CESM2, Rodgers et al. 2021). Inter-ensemble comparisons are conducted to estimate similarities and contrasts in the model forced responses, while the fidelity of their depictions of the energy budget, water cycle, and dynamical fields is assessed with the Climate Model Analysis Tool version 1 (CMATv1, Fasullo 2020). Their representations of a broad range of internal modes of variability are assessed in a companion manuscript (Fasullo et al. 2023).

# 2 Model and Ensemble Descriptions

## 2.1 The E3SM2 Large Ensemble

The techniques used to initialize LEs vary, with some LEs using a "micro" initialization in which the atmosphere state contains a small perturbation relative to other members, either consisting of a random roundoff-order perturbation or the selection of a slightly different time of initialization. In contrast and motivated by the desire to sample a broader diversity of ocean states, some ensembles employ a "macro" initialization in which multiple ocean states are chosen, typically to sample



a diversity of states of low-frequency modes. The E3SM2 LE adopts the macro approach, selecting initial years at decadal intervals in a prolonged preindustrial (PI) simulation.  This 21-member LE uses the historical (1850-2014) and future (2015-2100) SSP3-7.0 forcing protocols provided by the Coupled Model Intercomparison Project Phase 6 (CMIP6, Eyring et al., 2016). The model resolution is nominally 1° with 72 vertical levels for the atmosphere, 1° degree for the land, 0.5° for the river model, and variable resolution for the ocean and sea ice models which use a coarse grid in the midlatitudes (60 km) and

finer grids in the equatorial and polar regions (30 km). Improvements in model physics contribute to significant advances in the model's representation of clouds and precipitation versus E3SM1 (Golaz et al. 2022). To test the sensitivity of E3SM2 to the CMIP6 prescription of biomass burning emissions (vanMarle et al. 2017), an issue identified previously for CESM2 in Fasullo et al. (2022), an additional ensemble of 21 members is produced from approximately 1990 to 2085 using "smoothed" climatological satellite-era CMIP6 biomass emissions in a manner identical to that used for the CESM2 LE (Rodgers et al.

2021). A set of Detection and Attribution Model Intercomparison Project (DAMIP, Gillett et al., 2016) experiments is also used to isolate the responses to greenhouse gas and anthropogenic aerosol emissions.

## 2.2 The E3SM1 Large Ensemble

The E3SM1 LE is a 20-member ensemble from 1850 to 2100 that also uses the CMIP6 historical and SSP370 emissions pathways (Stevenson et al., 2023). The E3SM1 is the first version of the U.S. Department of Energy's Earth system model

(Golaz et al. 2019) and is designed to resolve resolutions relevant to energy applications (10's of km), though the LE is produced using a comparable resolution to the E3SM2 LE. The ensemble uses a macro initialization that samples a broad range of inter-basin ocean heat content states selected to span the distribution of variability in the Atlantic and Pacific basins. Details on this initialization strategy can be found in Stevenson et al. (2023). Only 17 members of the LE were available at the time of this work.

## 2.3 The CESM1 Large Ensemble

The CESM1 LE consists of 40 members that span from 1920 to 2100, using the CESM1 (Hurrell et al. 2013) initialized from a single member that spans 1850 to 2100 (Kay et al. 2015). The LE's micro-initialization approach generates inter-member contrasts through the imposition of round-off level perturbations to air temperature fields in 1920, with the coupled biogeochemical system spanning a broad range of internal states in the ensuing years. Produced in 2013, the ensemble uses

forcing estimates from phase 5 of the Coupled Model Intercomparison Project (CMIP5, Taylor et al. 2012) for both the historical and 21st centuries using the high forcing scenario RCP8.5 (Meinshausen et al. 2011). The model resolution is nominally 1° for all model components, similar to the CESM2 LE but with only 30 vertical levels in the atmosphere.

## 2.4 The CESM2 Large Ensemble

The CESM2 LE (Danabasoglu et al. 2020, Rodgers et al. 2021) consists of 100 members that span from 1850 to 2100. The

model resolution is nominally 1° for all model components with 32 atmospheric vertical levels and the LE uses both macro




and micro initializations. The first macro approach is used for 20 members of the LE based on start dates from the respective PI control simulation spaced at 10-year intervals. The second macro approach samples a maximum, minimum, and two transitional states of the Atlantic Meridional Overturning Circulation in the PI control simulation, with 20 micro ensemble members created for each of these four macro states using random perturbations of the atmospheric potential temperature field. During generation of the ensemble, a spurious warming arising from the CMIP6 prescription of biomass emission variability was identified, motivating the generation of 50 members using temporally smoothed biomass emissions (Fasullo et al. 2022). The E3SM2 smoothed biomass members already mentioned follow an identical approach as that used for the CESM2 LE (see Rodgers et al. 2021). A set of DAMIP experiments, analogous to those used for E3SM2, is also used to isolate the responses to greenhouse gas and anthropogenic aerosol emissions.

## 2.5 Observational Datasets

### 2.5.1 CERES Energy Balanced and Filled Radiative Fluxes

The satellite radiation data used here are from the CERES Energy Balanced and Filled (EBAF) Ed4.2 product (Loeb et al., 2018), which estimates monthly mean top-of-atmosphere (TOA) shortwave (SW), outgoing longwave (OLR), and net ($R_{TOA}$) radiative fluxes and solar irradiance measurements on a 1° grid from March 2000 through April 2023. TOA net solar radiation ($SW_{TOA}$) is determined from the difference between spatially and temporally averaged monthly solar irradiances and reflected SW fluxes. In comparison to simulated radiative fluxes, an issue arises from small differences in the atmospheric height at which SW, OLR, and $R_{TOA}$ are reported, which are typically at TOA for satellite retrievals and top-of-model (TOM) for simulations. In comparing $SW_{TOA}$ to SW flux at TOM ($SW_{TOM}$), however, we find distinctions between the fields shown in this work to be small, particularly in their changes over time (<0.1 W m$^{-2}$) and therefore the two levels are treated as equivalent.

### 2.5.2 Near-Surface Air Temperature Datasets

The observations of near-surface air temperature used in this work to evaluate historical-era trends are from the European Centre for Medium-Range Weather Forecasts (ECMWF) Twentieth Century (20C) Reanalysis (ERA20C; Poli et al. 2016) and the NOAA 20th Century Reanalysis Product (NOAA20C, Compo et al. 2011). These data are used as they extend through the 20C and are based on assimilated surface temperature information, infilling data gaps with model-estimated fields. Based on their contrasting methods in reconstructing climate, ERA20C is expected to perform better than NOAA20C in relatively well sampled regions such as western Europe, while 20CR is likely to account better for sampling gaps in regions such as the Southern Hemisphere middle to high latitudes as discussed in NCAR's Climate Data Guide (Poli and NCAR 2017).



### 2.5.3 The Climate Model Assessment Tool version 1 (CMATv1)

The CMATv1 is an objective analysis package for benchmarking coupled climate simulations through an evaluation against satellite and reanalysis datasets during the satellite era (Fasullo, 2020). The scoring system is designed to minimize susceptibility to internal variability and is based on pattern correlations of the mean state, seasonal contrasts, and El Niño/Southern Oscillation teleconnections. While all benchmarking approaches are based on a subjective selection of a finite number of metrics, and are therefore not wholly comprehensive, the value of CMATv1 stems from its use of dozens of feedback-relevant metrics (e.g. shortwave radiative fluxes, cloud radiative forcing) and a broad consideration of multiple fields and timescales. It is therefore one of the most comprehensive benchmarking packages available for coupled climate simulations. The influence of internal variability on its scoring metrics is also small and well-quantified, based on the CESM1 LE (Fasullo, 2020).

### 3 Large Ensemble Intercomparisons

Low frequency changes in the models' PI experiments are useful indicators of simulation drift, which results mainly from inconsistencies between the chosen initial ocean state and model physics. While the LEs used here are all well-balanced in the global mean for both near surface air temperature ($T_{2m}$ trend magnitudes <0.03 K c$^{-1}$, Fig. S1) and net radiation (mean $R_T$ magnitudes $\leq 0.12$ W m$^{-2}$, Fig. S2), regional drifts exist nonetheless. Drifts in the upper ocean (0-700 m, Fig. 1) and full-depth ocean (Fig. S3) are estimated from 70-year smoothed ocean heat content (OHC) anomalies (relative to the 20 years at the beginning of the interval shown). The drifts' magnitudes are important given their potential conflation with the FR. Though the global mean energetic imbalance in E3SM1 is modest (0.12 W m$^{-2}$), zonal-mean upper-ocean (0-700 m) drift is strong at many latitudes relative to other LEs examined here and it exhibits notable interhemispheric contrasts, with a cooling drift in the Northern Hemisphere (NH) and warming drift in the Southern Hemisphere (SH, Figs. 1a, S3a, S4a). Full-depth drift is similar in sign to the drift in the upper ocean but greater in magnitude, with strong opposing cooling and warming drifts in the NH and Southern Hemisphere (SH), respectively.

Upper-ocean drift in CESM1 is also strong at some latitudes, with features that include a cooling north of 45ºN and from 10ºS to 20ºN, and weak drift at most other latitudes (Fig. 1b). Drift in the full-depth ocean is characterized by cooling generally north of 10ºS and warming in the Southern Ocean (Fig. S3b). The sign of drift in the upper ocean in E3SM2 depends on latitude and is characterized generally by cooling in the Arctic and Tropics, and warming in the northern subtropics and Southern Ocean that largely offset each other in the global mean (Figs. 1c, S1, S2, S3c). At some latitudes, such as 40ºN, the trends are not monotonic, with amplitudes that vary in time and change in sign, and thus may instead be indicative of climate variability. Full-depth drift is characterized by cooling in the Tropics and midlatitudes, and warming at from 40ºN to 70ºN, though such changes are again not monotonic in time (Fig. S3c). In CESM2, upper-ocean drift is also spatially complex (Fig. 1d), with a cooling drift from approximately 20º-45ºN, with slight warming at most other latitudes



that grow over time. Drift in the full-depth ocean is characterized by a warming at nearly all latitudes that becomes particularly strong over time (Fig. S3d). The energy flux equivalents of these drifts, which are generally small, are shown in Fig. S4 to allow for comparison of drift magnitude to the radiative and energy flux responses shown in subsequent figures.

The effects of these drifts are removed in all subsequent analyses based on the linear trends computed from the models' PI experiments during the period of overlap with the plotted fields. In instances in which multiple initialization dates exist across the ensemble an average start date is used to define the period of overlap.

An analysis of global and hemispheric mean near surface air temperature ($T_{2m}$) from 1850 to 2100 is shown in Fig. 2, with an

analogous figure isolating historical era changes shown in Fig. S5. CESM1 has the coolest global mean $T_{2m}$ (286.3 K, Fig. 2a) during the base period (1920-50) while CESM2 has the warmest global mean $T_{2m}$ (287.2 K). Relative warmth across models in the PI simulation exhibit similar contrasts. Sufficient disagreement exists between the reanalysis datasets such that all models fall within the reanalysis range of base period $T_{2m}$ (286.3 to 287.4 K). That said, the E3SM1 and E3SM2 are conspicuous for their lack of warming during the second half of the 20C, a feature that is absent from both reanalyses and

CESM1/2. These biases and their drivers are addressed in Golaz et al. (2019, 2022) and further below, and are shown to be the likely result of excessive sulfate aerosol-driven cooling. By 2100, the E3SM1 LE warms more than the other LEs, in part due to its high climate sensitivity (Zheng et al. 2022).

Variability in $T_{2m}$ in the PI experiment is larger in CESM2 than in the other models, (Fig. 2a, left inset), a likely result of its

excessive ENSO variability (Fasullo et al. 2020). The NH is also considerably warmer in CESM2 during the base period and PI simulation than in the other models (Fig. 2b, left inset) though differences between the observations exceed 1 K, undermining definitive statements of model bias. Cooling in the 20C is particularly strong in the NH in E3SM1/2 (Figs. 2b, S5b). In addition to the effects of sulfate aerosols (Golaz et al. 2019, 2022, Zheng et al. 2022), drift is also a potential contributor to the lack of NH warming in E3SM1 (Figs. 1a). In the SH (Fig. 2c), CESM1 is about a degree cooler than the

other models, with both E3SM1/2 exhibiting a warm SH in the PI simulation (Fig. 2c, left inset). Warming in the SH in the late 20C in E3SM1/2 is also stronger than in the NH, though the SH warming is weaker than in reanalyses. The hemispheric gradient during the base period (1920-1950, Fig. 2d) is characterized by a NH that is warmer than the SH by about 1.5 K in reanalyses (values in parentheses). In the LEs this value varies greatly, as the NH is warmer than the SH in all cases but hemispheric contrasts are too weak in E3SM1/2 (0.6 K, 0.2 K), and too strong in CESM2 (2.1 K), as compared to reanalyses.


Imbalances in the energy budget are a key driver of the FR, and the net TOM flux ($R_T$, Fig. 3) is therefore a useful metric for assessing transient responses in the LEs. The E3SM and CESM LEs are generally in good balance during the PI, with absolute $R_T$ of $\leq 0.12$ W m$^{-2}$ (Fig. S2). As was the case for $T_{2m}$ (Fig. 2a), variability is greater in CESM2 in $R_T$ than in other models (Fig. 3a, left inset), suggesting the influence of excessive ENSO variance. A small but positive $R_T$ (heating) is

evident in all ensembles in the early 20C, with episodic intervals of cooling due to volcanic eruptions (Fig. 3a). Ensemble





mean $R_T$ in all LEs from 2000-2020 (values in parentheses) is less than in CERES (black line), whose value is 1.1 W m$^{-2}$, and $R_T$ is particularly small in E3SM2 (0.5 W m$^{-2}$). Trends in $R_T$ in CERES are also much larger than in any of the LE ensemble means. While the influence of internal variability may drive deviations greater than the ensemble-mean trend, only 7% and 5% of the members in the E3SM1 and CESM1 LEs, respectively, have trends as large as CERES and no E3SM2 or

CESM2 LE members exhibit trends as large, suggesting a contribution from errors in either prescribed forcings or model physics, as discussed further below.

The hemispheric energetic imbalance exerts an important influence on many aspects of climate and so both the hemispheric means and their contrasts are also assessed in Figure 3. Most volcanic eruptions exert a greater overall reduction in $R_T$ in the

NH due to their tendency to occur in the Tropics and NH, and asymmetries in the stratospheric circulation that enhance NH aerosol burdens even for tropical eruptions (Quaglia et al. 2023). This is evident for example in the transient signals in hemispheric differences, which are negative for most eruptions in the LEs, with the main exception being for the 1963 eruption of Mt. Agung in E3SM2 (Fig. 3d). Only CESM2 has a NH flux that is positive from 2000-2020 and, among the LEs, it agrees most closely with CERES. The existence of strongly negative NH $R_T$, particularly in E3SM2 (-1.9 W m$^{-2}$) may

relate to excessive aerosol forcing (Golaz et al. 2022) but is likely also influenced by structural model bias (e.g. in clouds), as similar inter-model contrasts are evident in the PI simulations (Fig. 3b). Conversely, all models except CESM2 simulate SH $R_T$ that is larger than observed (Fig. 3c), though CESM2 is also biased as it simulates values that are too small. Contrasts between hemispheres (NH-SH, Fig. 3d) in CESM2 are weaker than observed but are larger than the other LEs, which are too negative, particularly in E3SM1/2, an issue explored in depth in Golaz et al. 2019, 2022.


The time-latitude structure of warming is shown in Figure 4, and it exhibits many of the features anticipated from the global-scale time evolution of $R_T$ (Fig. 2). Common to the ensembles is a broadscale warming through 2100 that is greatest at high latitudes and is somewhat stronger in the Arctic than the Antarctic, consistent with the effects of Arctic amplification (Serreze et al. 2011). An additional feature of E3SM1/2 that is not evident in CESM1/2 is the strong 20C cooling evident

from 30º-70ºN (Figs. 4a,c), addressed in both Golaz et al. (2019, 2022) and Zheng et al. (2022), and attributed to an excessive cooling response to anthropogenic sulfate aerosols. Time series from single-forcing experiments support this interpretation, as the aerosol response in $T_{2m}$ is found to be about twice as large in E3SM2 as in CESM2 (Fig. S6). Simulation drift is also a likely contributor to mid 20C NH midlatitude cooling (Figs. 1, S3). The aerosol cooling signal is the first FR that emerges from the noise of internal variability in both E3SM1 (lack of stippling where significant in all

figures) and E3SM2. In CESM1, the identification of emergent signals and differences with CESM2 prior to 1920 is not possible due largely to the availability of only a single ensemble member (stippling before 1920 in Fig. 4b/d). Instead, Arctic warming is the first forced response to emerge, which occurs shortly after the initialization of the ensemble in 1920 (Fig. 4b). Though somewhat delayed versus E3SM1, 20C NH cooling in E3SM2 is stronger than in E3SM1 at most times and latitudes, particularly in the Arctic (as evident from the lack of stippling from 1920-2000 from 60-90º N in Fig. 4c).





Warming in the mid to late 21st Century (21C) is greater in E3SM1, which has an unrealistically large equilibrium climate sensitivity (Golaz et al. 2019). For CESM1/2, the large number of ensemble members after 1920 increase the detectability of intergenerational differences (lack of stippling in most regions of Fig. 4d). Though the general patterns of warming are similar, some differences are evident such as the elevated future warming from 0-20ºS in CESM2. Warming above 5K in the NH also extends farther south in E3SM than in CESM. Comparison between CESM1 and the other models is complicated

however by contrasts in prescribed climate forcings, with CESM1 using RCP85 and other LEs using SSP370.

The time-latitude evolution of $R_T$ is a key indicator of the influence of forcing and is shown in Figure 5. In E3SM1/2, the 20C evolution is characterized by robust negative $R_T$ anomalies (cooling) that begin in the late 19th century from 30º-70ºN, and that intensify into the late-20C in conjunction with positive anomalies (heating) that emerge and intensify in the low

latitude SH (indicated by lack of stippling in Fig. 5a). Analysis of precipitation (to be discussed below in Fig. 8) shows the SH features to be related to displacements of tropical deep convection, consistent with the response to sustained NH cooling (Hwang and Frierson 2013). While locations and timings of mid 20C forced $R_T$ anomalies in CESM1/2 similar to those E3SMv1/2 are evident (e.g. lack of stippling in Fig. 5b), their magnitudes are weaker. Short-lived cooling pulses across a broad range of latitudes are also evident in all LEs and these are driven by major volcanic eruptions. In the 21C, the

latitudinal structures of $R_T$ anomalies exhibit common features across the ensembles, including a broad-scale heating that is greatest in the Arctic, and a heating-cooling dipole south of 60ºS. Other details in the structure, such as trends between 10ºS and 40ºS, are strongly model dependent and likely relate to cloud responses to warming and adjustments to $CO_2$, such as for example the rapid SH subtropical cloud adjustment to $CO_2$ in CESM2 (Fasullo and Richter, 2023). Detectable differences between successive model generations are also evident at various times and latitudes (lack of stippling in Figs. 5c, d). In

CESM however, the interpretation of such differences is complicated by the potential role for contrasts in the forcing scenarios used for both the historical and future eras and therefore cannot be directly attributed to model version (Fasullo and Richter 2023).

As dominant contributors to anomalies in $R_T$ and their differences across models, changes in $SW_{TOM}$ highlight important

contrasts across the LEs. The time-latitude structure of $SW_{TOM}$ anomalies is shown in Figure 6. In E3SM1/2 (Fig. 6a, c), the 20C evolution is characterized by robust cooling anomalies that begin in the late 19C from 30º-70ºN that intensify into the late-20C, similar to anomalies in $R_T$. Unlike $R_T$ anomalies however, there is little change in $SW_{TOM}$ in the SH during the mid 20C, suggesting a role for high clouds and reduced longwave fluxes tied to changes in deep convection in dictating changes in $R_T$ (Fig. 5). While episodes of negative forced anomalies in CESM1/2 in the 20C are evident (e.g. lack of stippling in Fig.

6b), they are shorter-lived and their magnitudes are significantly weaker than in E3SM1/2. An influence of volcanic eruptions is again evident in the episodic cooling pulses in the 20C in all LEs (across many latitudes). In the 21C, the latitudinal structure of $SW_{TOM}$ anomalies exhibit common features across the ensembles, such as a broad-scale heating that is



evident in the extratropics in all ensembles except at 60ºS, where at times the signs of model trends disagree. Contrasts in the timing and magnitudes of projected changes are also evident across latitudes.


The effects of forced responses, such as the NH cooling in the mid to late 20C and global warming in the 21C, extend beyond temperature and include the water cycle due in part to the energetic linkages between these fields (Trenberth et al. 2009). Responses in the LEs in precipitable water (PrW), which is the integrated water vapor in the atmosphere expressed in liquid equivalent depth, are shown in Figure 7. With cooling, the capacity of air to hold moisture decreases and forced

reductions in PrW are therefore coincident in E3SM1/2 with periods of cooling across the NH in the mid 20C. Forced reductions in E3SM1 (Fig. 7a) are first simulated in the late 19C (coincident with the eruption of Krakatoa in 1883) and persist through the 20C, reaching a peak intensity near 1 mm in the 1960s and 1970s. Reductions of similar intensity and timing are evident in E3SM2 and the PrW increases in the SH are coincident with enhancement of tropical precipitation (to be discussed further below). Responses in the 20C are small however relative to projected increases in PrW in association

with projected warming (Fig. 4), with increases that exceed 8 mm in the Tropics and subtropics in all LEs by the late 21C. Increases in PrW in CESM1/2 are first evident in the SH in the mid 20C. In the 21C, increases are approximately symmetric about the equator in E3SM1 and CESM1/2 but are skewed toward the NH in E3SM2, where the greatest increases are located north of 20ºS, consistent with the somewhat muted warming in E3SM2 (Fig. 4c) and a fixed relative humidity constraint. Increases south of 70ºS are relatively small in all LEs, likely due to limitations on surface water availability and

very low mean state temperatures and PrW values over Antarctica.

The water cycle perturbations responses in PrW are associated with spatially complex responses in precipitation (P), shown in Figure 8. With a cooler lower atmosphere (Fig. 4), less SW flux available at the surface to supply the energy consumed by evaporation (Fig. 6), and reduced PrW (Fig. 7), the NH climate in E3SM1/2 experiences significant forced reductions in P

across the 20C at all latitudes (Figs. 8a, c). Also, the southward shift in deep convection in E3SM1/2, cited above, is expressed as decreases in P in the Tropics, and increases in P from 5ºS to 20ºS that peak in the 1970s. The spatial structure of anomalies during this time is characterized by particularly strong reductions in P in the western Pacific warm pool and NH deep convective regions and increases south of the equator across much of the SH (not shown). Similar responses in P in CESM1/2 also emerge from background variability (Figs. 8b, d) but are weaker at most latitudes, particularly in CESM2,

and do not extend as far north as in E3SM1/2. Projected changes are characterized by robust increases in P in all LEs on the equator and in the middle to high latitudes while decreases are projected in the subtropics generally, though with magnitudes, latitudinal bounds, and timings that vary across LEs.

Meridional atmospheric heat transports (MHT$_{atm}$) are strongly coupled to the latitudinal structures of thermal and moisture

fields and their forced changes are shown in Figure 9. In E3SM1/2, increases in MHT$_{atm}$ are evident north of 20ºS in the 20C, which are particularly strong (>0.1 PW) from 1960-2000 and coincide with strong aerosol-induced cooling (Fig. 4-6).



The initial emergence of forced increases in E3SM occurs in the late 19th C. The increased meridional thermal gradient arising from aerosol forcing is a likely contributor to the mid to late 20C $MHT_{atm}$ maximum (Needham et al. 2023). Changes in CESM during the 20C are weak compared to those in E3SM, with increases near 0.08 PW at low latitudes. In the 21C,

changes are characterized by increased poleward transport of order 0.2 PW, characterized by positive (negative) $MHT_{atm}$ in the NH (SH), but with strong hemispheric and model dependence. Increases in $MHT_{atm}$ in the NH are weak in E3SM1 and largely absent from E3SM2 (Fig. 9a, c), likely due to the disproportionately strong 21C surface warming in the NH (Fig. 4) and the associated weakening of the meridional temperature gradient. Projected $MHT_{atm}$ increases in the NH are particularly pronounced in CESM2 and are, in part, associated with the large projected increase in $R_T$ and $SW_{TOM}$ near 20ºS which

contributes to increased low latitude atmospheric energy divergence (Fig. 6d).

Meridional oceanic heat transports ($MHT_{ocn}$) exert an influence that is generally strongest equatorward of 30ºN/S in the climatological mean (Trenberth and Fasullo, 2017) and their forced changes are shown in Figure 10. In E3SM1/2, increases in $MHT_{ocn}$ are evident north of 20ºS in the 20C, which are particularly strong (>0.2 PW) from 1960-2000 and, as with

$MHT_{atm}$, coincide with strong aerosol-induced cooling (Fig. 4-6). Changes in CESM1/2 in the 20C are relatively weak, with increases near 0.05 PW from 1960-2000. In the 21C, forced reductions in poleward $MHT_{ocn}$ are evident in all LEs, but with strong model dependence and large magnitudes in CESM and particularly in CESM1. Projected decreases in the NH are likely tied to changes in the Atlantic Meridional Overturning Circulation (AMOC) and the lack of strong NH decreases in E3SMv1/2 may reflect weak AMOC conditions in the present-day (Hu et al. 2020) and the associated limited potential for

future weakening.

As the ocean stores over 90% of Earth's energy imbalance, model-dependence in climate system storage is reflected in contrasts in ocean heat content (OHC) trends and these are shown for the surface to 2000 m depth in Figure 11. Changes in OHC are small in the 19C, though CESM1 and E3SM1 exhibit detectible cooling by 1900 and are notably cooler than

CESM2 and E3SM2 by 1950. In E3SM1 and E3SM2, the evolution of OHC after 1950 are quite different than in CESM, with strong cooling through the late 20C, consistent with the aerosol effects already identified. Significant contrasts between models are also evident in the 21C, with OHC increases in CESM1/2 being significantly greater than in E3SM1/2. The weak heat uptake in E3SM1/2, despite being associated with comparable surface warming (e.g. Figs. 2, 4), is likely to be linked with a weak AMOC in the models, with the effect of decreasing heat uptake by the deep ocean, consistent with the findings

of Hu et al. (2020) for E3SM1 which linked the model's high transient climate response to weakness in AMOC. Though TCR decreased in E3SM2 from E3SM1, it remains much larger than in either CESM1 or CESM2. This lack of ocean heat uptake in E3SM1 and E3SM2 may in turn contribute to strong 21C NH warming (Fig. 4) and small changes in $MHT_{atm}$ (Fig. 9).



## 4 Benchmarking

Summary scores for the model benchmarking tool CMATv1 (Fasullo 2020), which compares various aspects of simulated dynamic, energy budget, and water cycle fields to satellite and reanalysis estimates, are shown in Table 1. Scores based on spatial pattern correlations for the overall climate and for targeted fields and timescales are shown, where larger magnitude values represent greater skill in representing the respective fields. In the CMATv1 design, internal variability in the benchmarking metrics is designed to achieve specific thresholds based on analysis of the CESM1 LE. The scores provide a

range of insights into inter-model and inter-generational differences in the LEs and their significance, something also demonstrated for the CMIP ensembles in Fasullo (2020), where progressive improvement across model generations is identified. First, E3SM1 is generally the lowest scoring model of the four, both in terms of the overall score (0.776) and more targeted scores in Table 1. E3SM1 scores particularly poorly in depicting ENSO teleconnections (0.583). Major improvements in E3SM2 from E3SM1 are apparent in the energy budget (from 0.782 to 0.821) and water cycle scores (from

0.745 to 0.767), and for ENSO teleconnections (from 0.583 to 0.653), which is the highest of the LEs assessed here (though within the uncertainty ranges of both CESM1 and CESM2). Scores for other summary metrics are highest for CESM2 and its improvements from CESM1 are evident in all metrics.

To illustrate examples of simulated biases relevant to the CMATv1 benchmarks in Table 1, and the differences between

E3SM1/2 and CESM1/2, biases in annual mean $R_T$ are shown in Figure 12. The biases are important as they influence the spatial gradients of temperature and moisture and thereby impact dynamics and MHT. Biases in E3SM1 and CESM1 are widespread in tropical and NH ocean regions, with $R_T$ that is too small. Exceptions include the regions of stratocumulus cloud decks west of Mexico and Peru, where $R_T$ is generally too large due to excess SW absorption and associated with deficient stratocumulus cloud decks (not shown). Over land, $R_T$ biases are generally positive, except in equatorial Africa,

southern India, and South America, where it is often biased low, particularly in CESM1. Low biases are evident in the Tibetan Plateau in E3SM1 and E3SM2, which are not evident in CESM. The lowest root-mean-squared error (RMSE) in $R_T$ is found for CESM2 (7.8 W m$^{-2}$), where regional biases are smaller than in CESM1 and E3SM1/2, while the highest RMSE is found for CESM1 (11.2 W m$^{-2}$). The location of widespread ocean biases in CESM2 has also shifted to be largest near 50ºS, where it is underestimated, while the other models tend to overestimate $R_T$ in the region.


Biases in precipitation (P) identified in CMATv1 for climatological mean fields from 1979 to 2020, based on comparison against GPCP, are shown in Figure 13. In all ensembles a common pattern of biases exists, characterized by excess P in the off-equatorial Pacific Ocean (characteristic of the ubiquitous double ITCZ issue) and the western Pacific warm pool, and deficient P in the equatorial Pacific Ocean and over much of South America. Pattern correlations improve slightly from

versions 1 to 2 of both E3SM (0.88 to 0.85) and CESM (0.85 to 0.89) and RMSE is lowest for E3SM2 and CESM2 (0.99), due largely to reduced biases in the south-eastern subtropical Pacific Ocean.



## 5 Sensitivity to CMIP6 Biomass Emissions

Finally, the sensitivity of E3SM2 to CMIP6 prescribed emissions is explored in Figure 14. In Fasullo et al. (2022), a

sensitivity in CESM2 to these emissions was shown to drive a strong high latitude warming, owing to an abrupt increase in emission variability in 1997 that via nonlinear interactions with clouds drove a rectified reduction in mean albedo from 40ºN -70ºN. Here, based on the ensemble mean differences between the E3SM2 LE and smoothed biomass LE it is shown further that E3SM2 exhibits a similar, albeit somewhat weaker, response. The response is characterized for example by reductions in cloud albedo (Fig. 14a) and increases in $T_{2m}$ (Fig. 14b), $SW_{TOM}$ (Fig. 14c), and surface net SW flux ($SW_{SFC}$ Fig. 14d), though

with magnitudes that are reduced somewhat from those in CESM2 (dashed). These net reductions correspond to extremes in biomass emissions, which are particularly high in 1998 and 2003 and relatively low in most other years (see Fasullo et al. 2022, Figure 1f). These variations result in radiation and $T_{2m}$ anomalies that are negative during years of high emissions but positive and of comparable magnitude during the more frequent years of low emissions, and thus drive a net warming. The spatial structure of the warming (Fig. 14e) is characterized by the strongest responses over NH land and the Arctic Ocean,

where a warming response up to 0.5K is simulated. Details of the interactions between emissions, clouds, radiation, and the broader climate state will be addressed in follow-on work.

## 6 Conclusions

The unique value of LEs, which includes the opportunity to estimate forced climate responses and make robust comparisons

across models, is illustrated in this work. In doing so, the LEs provide estimates of the potentially predictable component of the climate response arising from changes in its external forcings, which include most prominently industrial sulfate aerosols in the 20C and greenhouse gasses in the 20C and 21C, and allow for an assessment of inter-model contrasts. Understanding these structural uncertainties provides insight for interpreting historical-era changes in nature and for quantifying the range of plausible 21C climate outcomes, the factors underlying their differences, and associated uncertainties in a changing

climate.

In this work, four recently produced LEs are intercompared and assessed with reanalysis and satellite datasets. The analysis summarizes many features of agreement in simulated climate across the LEs, which include a mid-20C cooling driven by aerosols and an associated water-cycle response, a polar amplification of warming and associated albedo reductions, increases in PW across latitudes, and latitudinally complex changes in P. Areas of disagreement across the LEs arising from

contrasts in both model structure and imposed forcings, include contrasts in the magnitudes of mid to late 20C cooling, the structure of associated low-latitude P responses, and changes in MHT. The contrast in climate forcings used in CESM1 versus the other LEs limits strict statements regarding some of the comparisons made, both for historical (e.g. smoothed biomass) and future climates, and highlights the uncertainties associated with climate forcing agents (Fyfe et al. 2021, Holland et al. 2023).



In benchmarking the ensembles, robust improvements in E3SM and CESM are identified in the progression from versions 1 to 2. These improvements are particularly large in the energy budget and water cycles of E3SM, and in its simulated ENSO teleconnections. The analysis also identifies a sensitivity in E3SM2 to the variable nature of CMIP6 biomass emissions similar to, but somewhat weaker than, that identified in CESM2 in prior work. Caution should therefore be exercised in evaluating transient climate features of the satellite era in both CESM2 and E3SM2. The failure of E3SM1 and E3SM2 to

adequately warm during the late 20C is also found to be a major shortcoming of the ensemble, with impacts on their simulation of the water cycle, and this feature is attributed to the models' excessive sensitivity to industrial sulfate aerosols and, with a secondary contribution from model drift. A notable interhemispheric contrast in drift is also identified for E3SM1. Lastly, even during the early 21C, very few LE members are found to exhibit trends in $R_T$ as large as observed from CERES, thus motivating further study on the origin of this apparent disagreement. Lastly, it is also noted that despite both

being high scoring models, E3SM2 and CESM2 project very different forced responses of radiation, precipitation, and meridional heat transport in both the atmosphere and ocean, underscoring the challenges that exist in narrowing future projections from evaluation with present-day observations alone.

**Acknowledgements**

Portions of this study were supported by the Regional and Global Model Analysis (RGMA) component of the Earth and Environmental System Modeling Program of the U.S. Department of Energy's Office of Biological & Environmental Research (BER) under Award Number DE-SC0022070, and by the National Center for Atmospheric Research, which is a major facility sponsored by the National Science Foundation (NSF) under Cooperative Agreement No. 1852977. The development of the E3SM model is supported by the E3SM project funded by the Office of Biological and Environmental

Research in the US Department of Energy's Office of Science. The efforts of Dr. Fasullo in this work were also supported by NASA Awards 80NSSC17K0565 and 80NSSC22K0046, and by NSF Award 2103843. Work at LLNL was performed under the auspices of the U.S. Department of Energy by Lawrence Livermore National Laboratory under Contract DE-AC52-07NA27344. This research used resources of the National Energy Research Scientific Computing Center (NERSC), a DOE Office of Science User Facility supported by the Office of Science of the U.S. Department of Energy under Contract No.

DE-AC02-05CH11231 using NERSC award ALCC-ERCAP0022631.



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



**Tables**

**Table 1: CMATv1 summary metrics (Fasullo 2020) for E3SM1, CESM1, E3SM2, CESM2 ensembles with twice the ensemble standard error indicated. Overall scores differing by 0.01 for the ensembles exceed the likely influence of internal variability.**

|  | Overall | Energy | Water | Dynamic | Mean | Annual Cycle | ENSO |
|---|---|---|---|---|---|---|---|
| E3SM1 | 0.776±0.008 | 0.782±0.009 | 0.745±0.008 | 0.802±0.009 | 0.875±0.001 | 0.874±0.002 | 0.583±0.023 |
| CESM1 | 0.803±0.004 | 0.809±0.004 | 0.762±0.004 | 0.839±0.004 | 0.889±0.001 | 0.887±0.000 | 0.640±0.011 |
| E3SM2 | 0.801±0.008 | 0.821±0.008 | 0.767±0.008 | 0.816±0.009 | 0.885±0.001 | 0.873±0.001 | 0.653±0.024 |
| CESM2 | 0.814±0.004 | 0.827±0.003 | 0.772±0.004 | 0.843±0.004 | 0.909±0.001 | 0.893±0.001 | 0.647±0.010 |



## Figures



**Figure 1: Time-space evolution of ocean heat content changes in the preindustrial simulations for the top 700 m in the (a) E3SM1, (b) CESM1, (c) E3SM2, and (d) CESM2, respectively, after the approximate time of the ensemble initialization, which in some cases varies by ensemble member. Time intervals shown are chosen to correspond to 1850 to 1990 in the historical era. In cases of variable initialization dates, an approximate date range is chosen (years 1000 for CESM2, 200 for E3SM1, and 100 for E3SM2). A 70-year running smoothing is applied to reduce internal variability.**



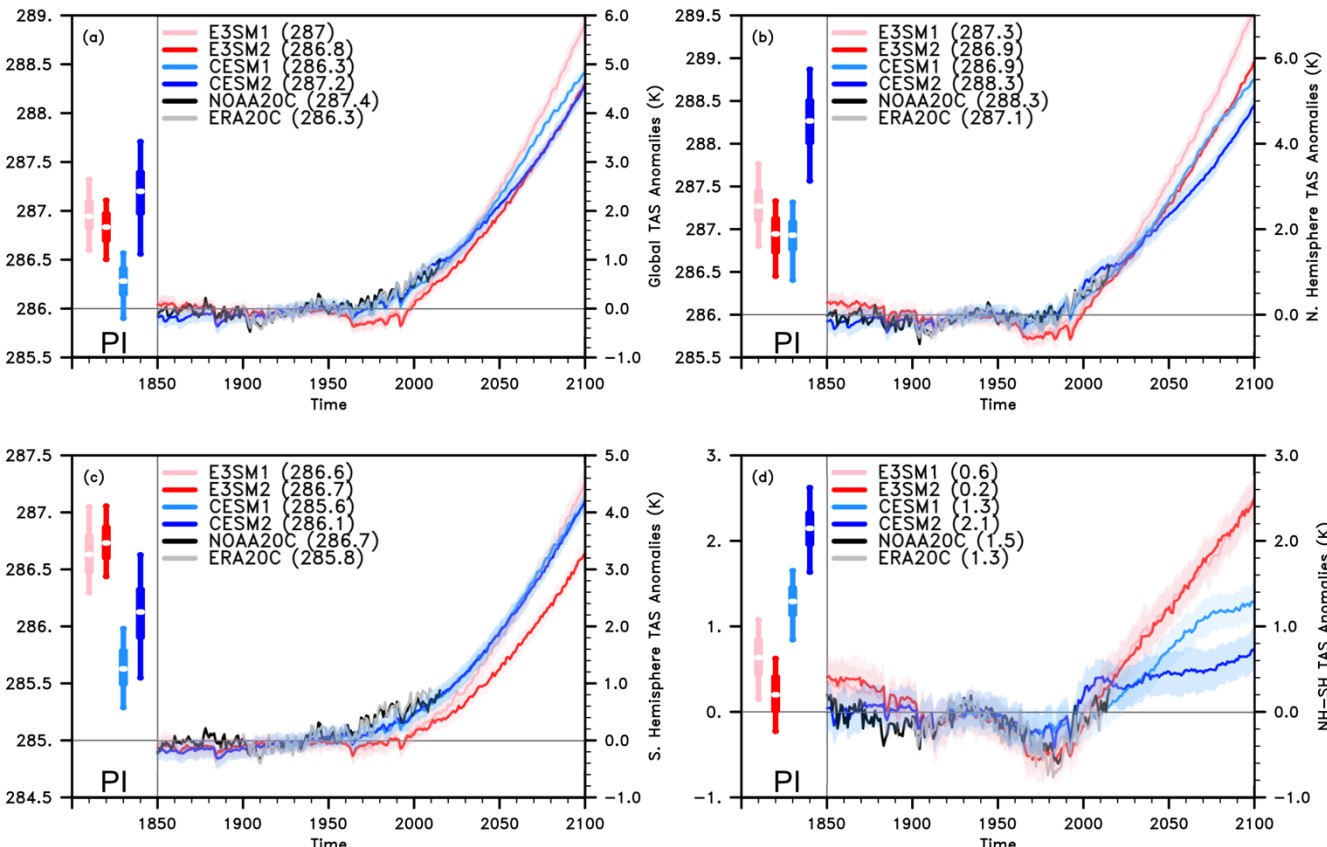

**Figure 2: Evolution of mean near surface air temperature anomalies (K) for E3SM1/2 and CESM1/2 for (a) the globe, (b) NH, (c)**
**SH, (d) NH-SH, with drifts removed. The minimum, maximum, median, and interquartile range of annual means in the**
**preindustrial simulations are also shown, left axes. Observation-based estimates from NOAA20C (black) and ERA20C (grey) are**
**indicated. A base period of 1920-50 is used and its values for each region are indicated in parentheses. All ensembles use the future**
**SSP-3.70 scenario except CESM1 which uses RCP8.5.**




**Figure 3: Evolution of top-of-model net radiative flux (W m⁻²) for E3SM1/2, CESM1/2, and observations for 2000-2022 from CERES for (a) the globe, (b) NH, (c) SH, (d) NH-SH, with drifts removed. The minimum, maximum, median, and interquartile range of annual means in the preindustrial simulations are also shown, left axes. Values from CERES (black) are also shown. All ensembles use the future SSP-3.70 scenario except CESM1 which uses RCP8.5.**






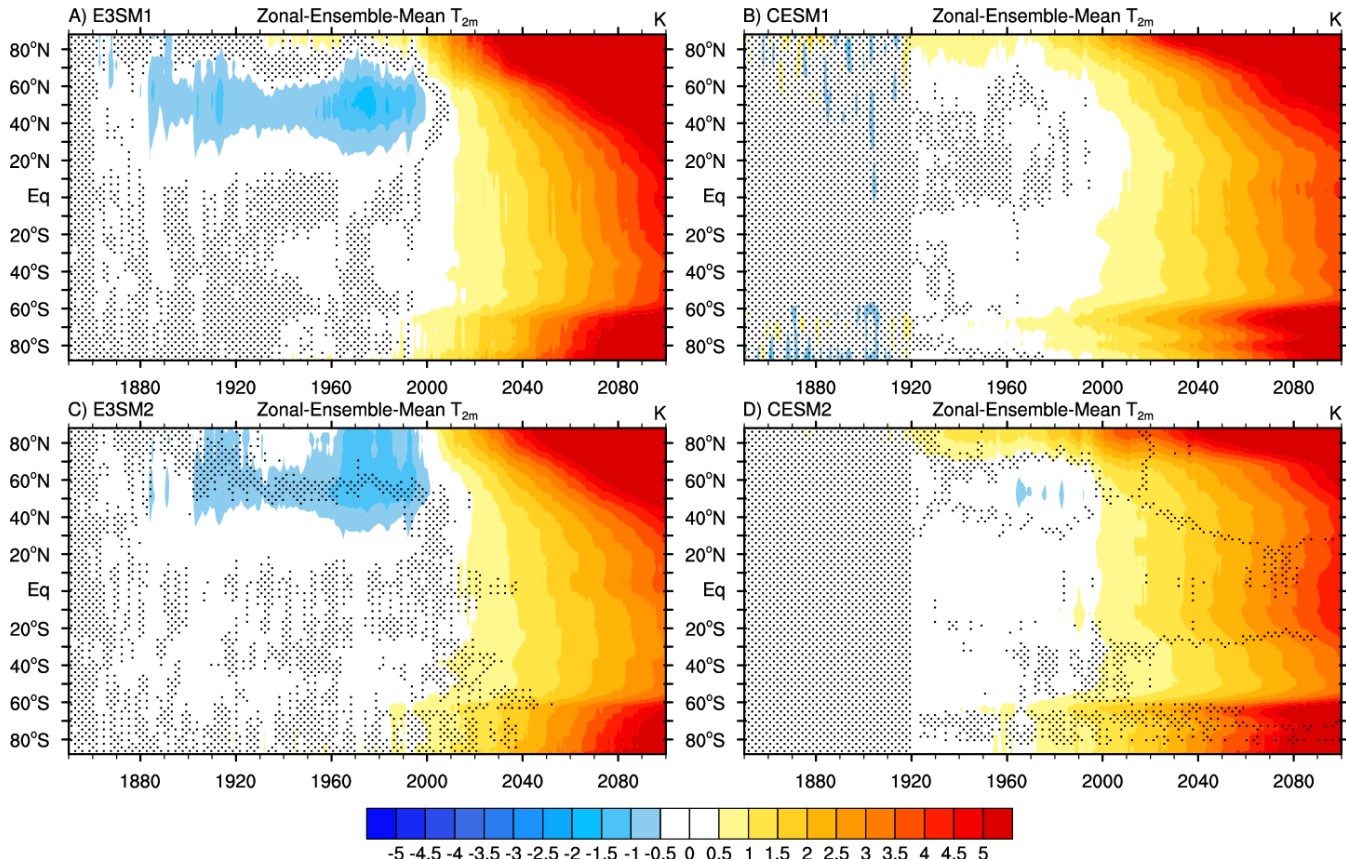

**Figure 4: Ensemble-mean change in two-meter air temperature (K) from the 1850-59 average in E3SM1 (a) CESM1 (b) E3SM2 (c), and CESM2 (d), with drifts removed. Stippling indicates changes less than twice the standard error in (a, b) and inter-generational differences (e.g. E3SM1 versus E3SM2) less than twice the standard error in (c, d). All ensembles use the future SSP-3.70 scenario except CESM1 which uses RCP8.5.**




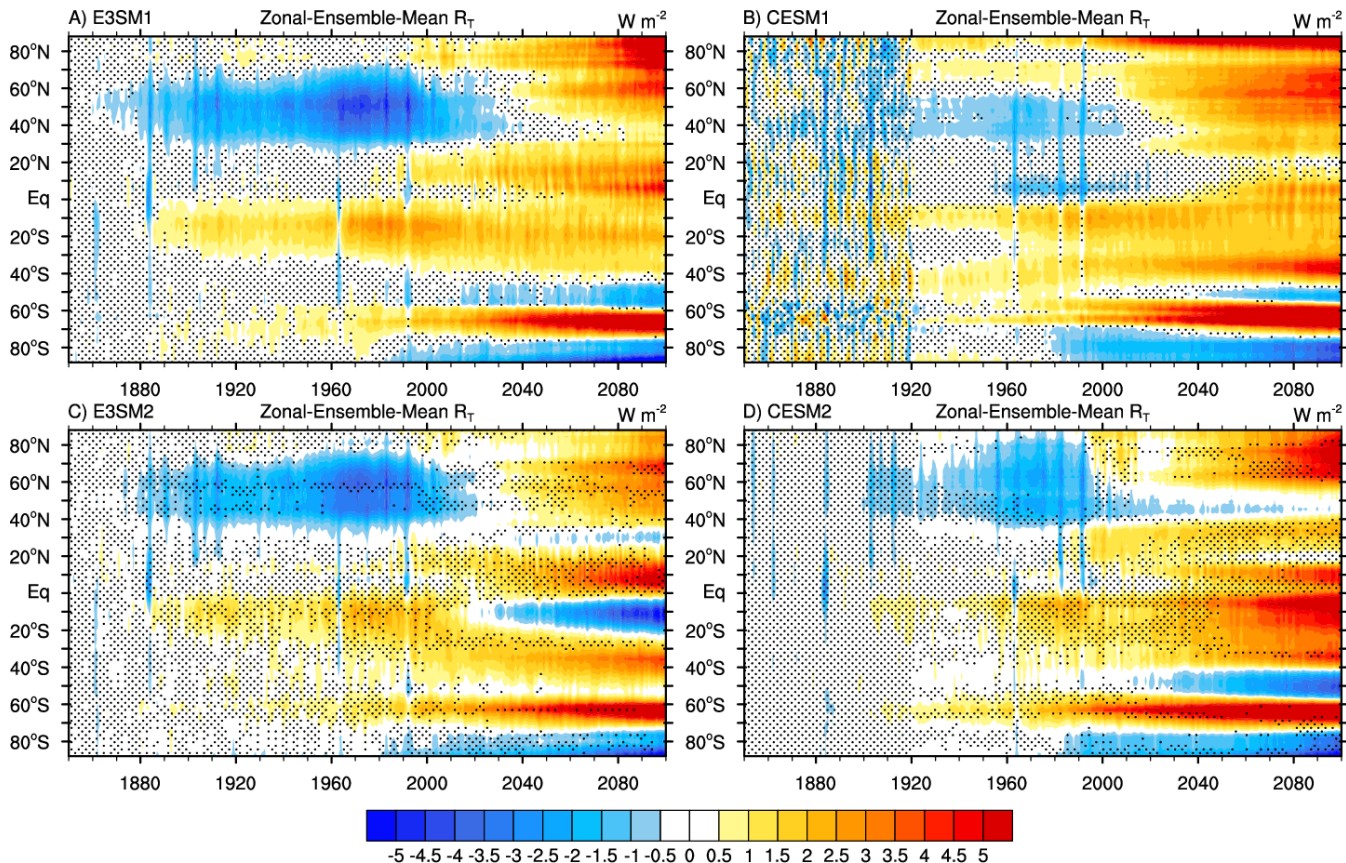


**Figure 5: Ensemble-mean change in net top-of-model radiation (W m⁻²) from the 1850-59 average in E3SM1 (a) CESM1 (b) E3SM2 (c), and CESM2 (d), with drifts removed. Stippling indicates changes less than twice the standard error in (a, b) and intergenerational differences less than twice the standard error in (c, d). All ensembles use the future SSP-3.70 scenario except CESM1 which uses RCP8.5.**






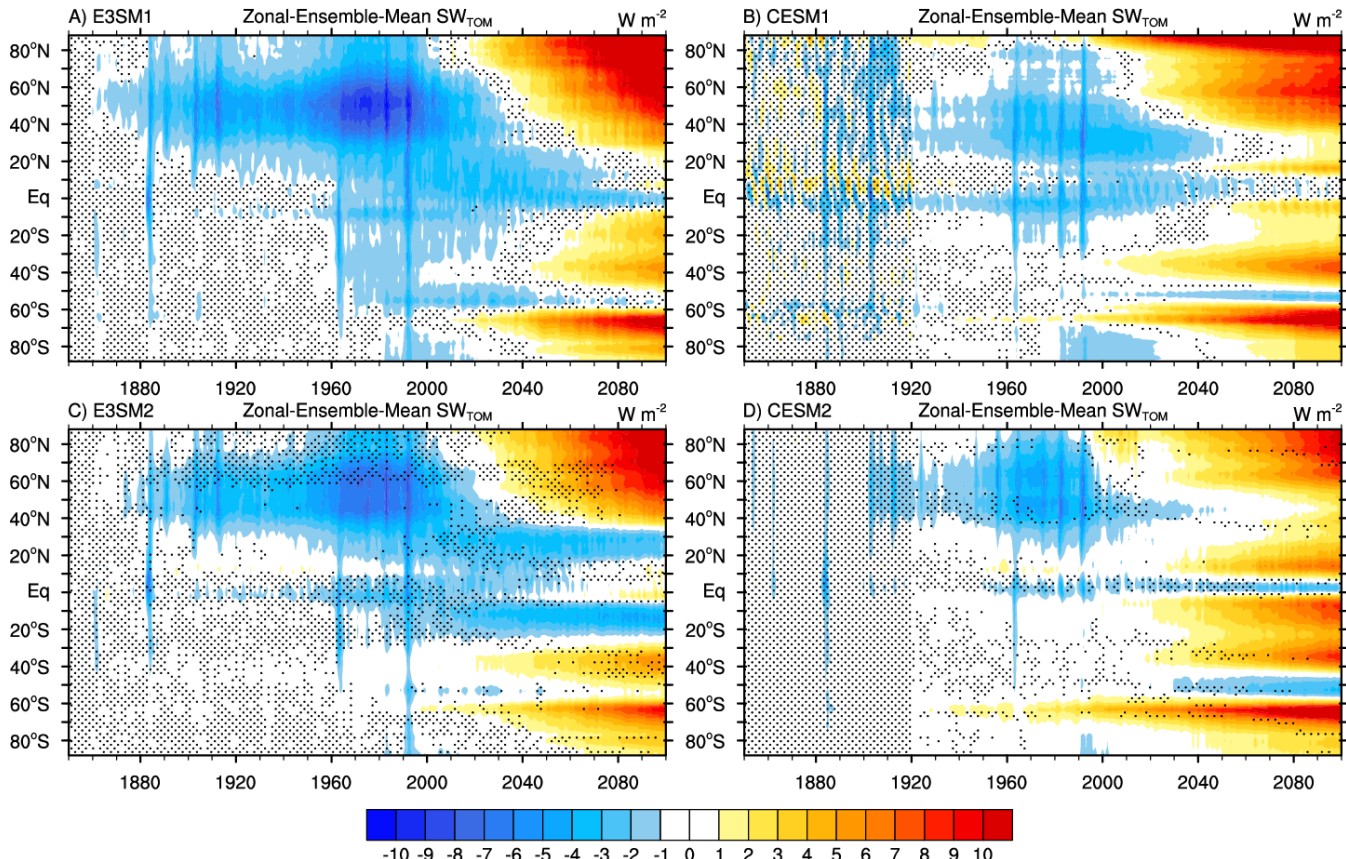

Figure 6: Ensemble-mean change in net top-of-model absorbed shortwave radiation (W m$^{-2}$) from the 1850-59 average in E3SM1 (a) CESM1 (b) E3SM2 (c), and CESM2 (d), with drifts removed. Stippling indicates changes less than twice the standard error in (a, b) and intergenerational differences less than twice the standard error in (c, d). All ensembles use the future SSP-3.70 scenario except CESM1 which uses RCP8.5.



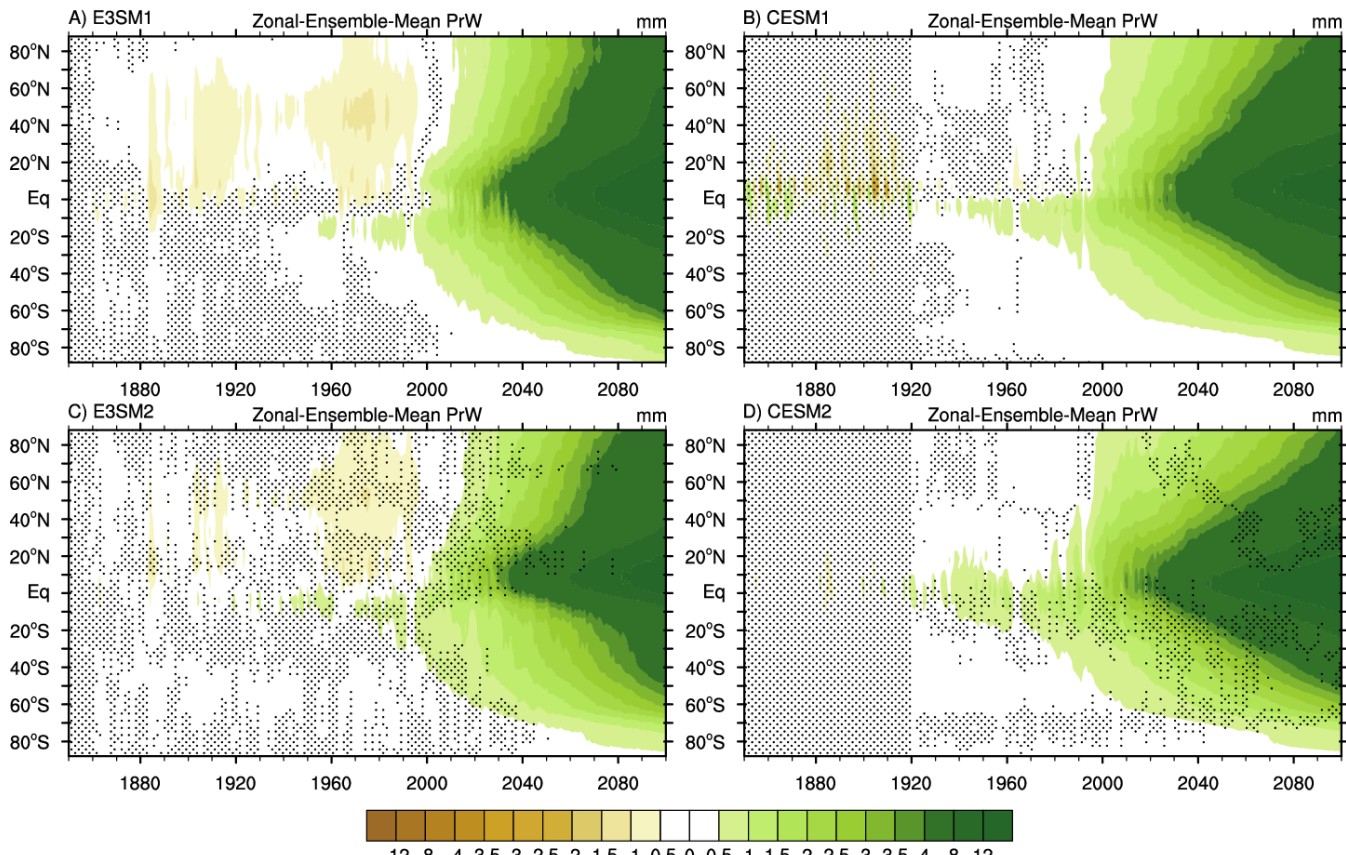

**Figure 7: Ensemble-mean change in precipitable water (mm) from the 1850-59 average in E3SM1 (a) CESM1 (b) E3SM2 (c), and CESM2 (d), with drifts removed. Stippling indicates changes less than twice the standard error in (a, b) and intergenerational differences less than twice the standard error in (c, d). All ensembles use the future SSP-3.70 scenario except CESM1 which uses RCP8.5.**




**Figure 8: Ensemble-mean change in precipitation (mm day⁻¹) from the 1850-59 average in E3SM1 (a) CESM1 (b) E3SM2 (c), and CESM2 (d), with drifts removed. Stippling indicates changes less than twice the standard error in (a, b) and intergenerational differences less than twice the standard error in (c, d). All ensembles use the future SSP-3.70 scenario except CESM1 which uses RCP8.5.**





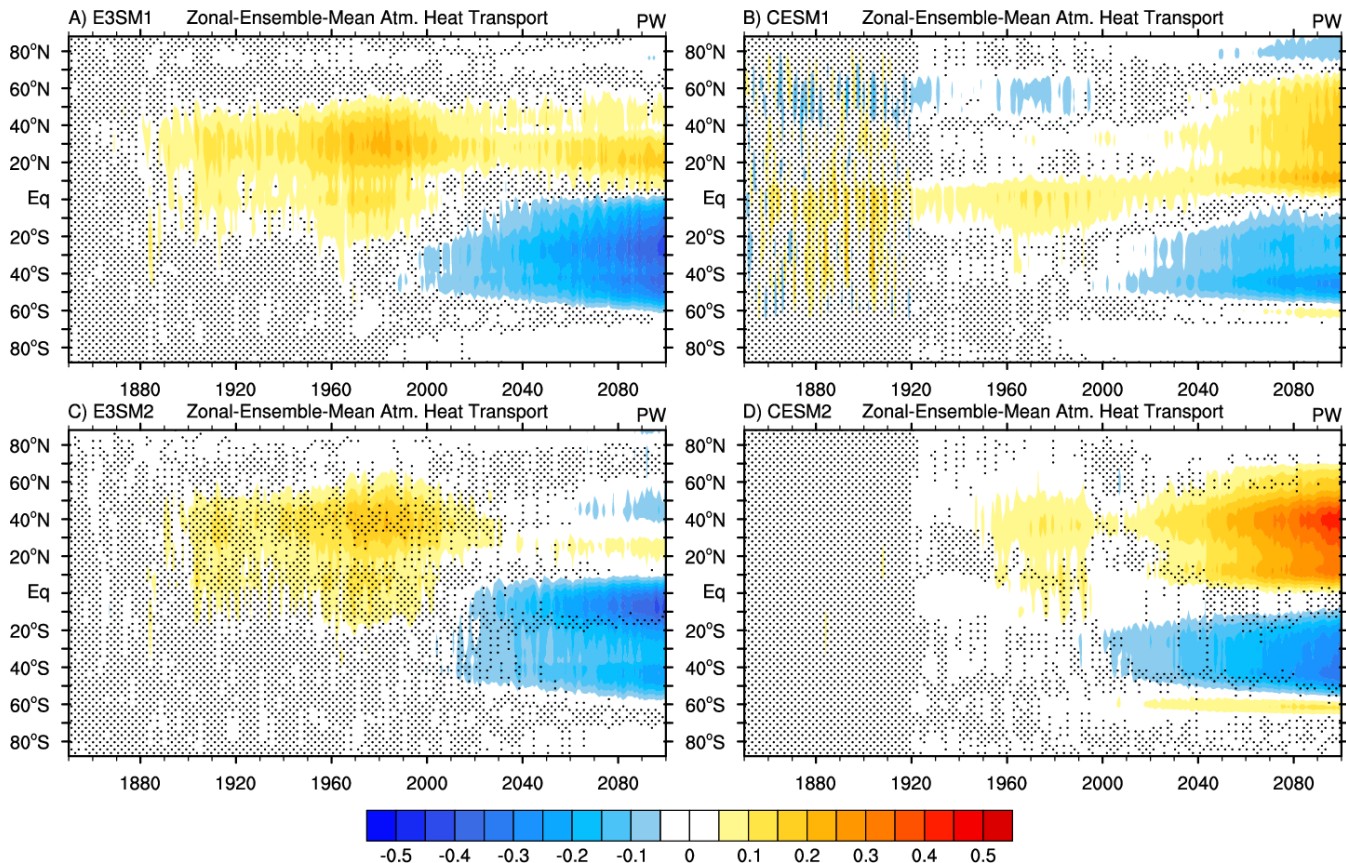

Figure 9: Ensemble-mean change in meridional atmospheric heat transport (PW) from the 1850-59 average in E3SM1 (a) CESM1 (b) E3SM2 (c), and CESM2 (d), with drifts removed. Stippling indicates changes less than twice the standard error in (a, b) and intergenerational differences less than twice the standard error in (c, d). All ensembles use the future SSP-3.70 scenario except CESM1 which uses RCP8.5.



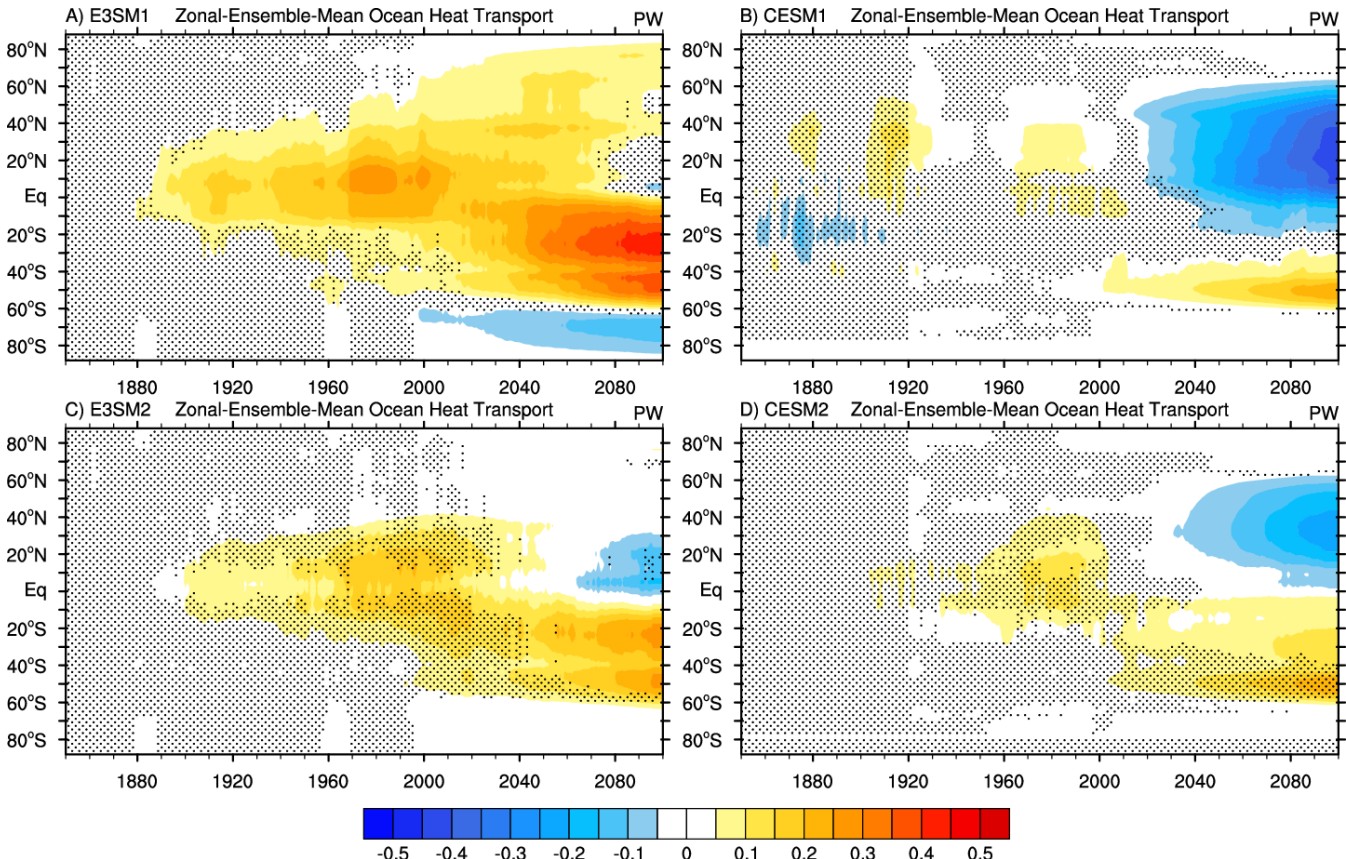

**Figure 10: Ensemble-mean change in meridional ocean heat transport ($10^{15}$ W) from the 1850-59 average in E3SM1 (a) CESM1 (b) E3SM2 (c), and CESM2 (d), with drifts removed. Stippling indicates changes less than twice the standard error in (a, b) and intergenerational differences less than twice the standard error in (c, d). All ensembles use the future SSP-3.70 scenario except CESM1 which uses RCP8.5.**



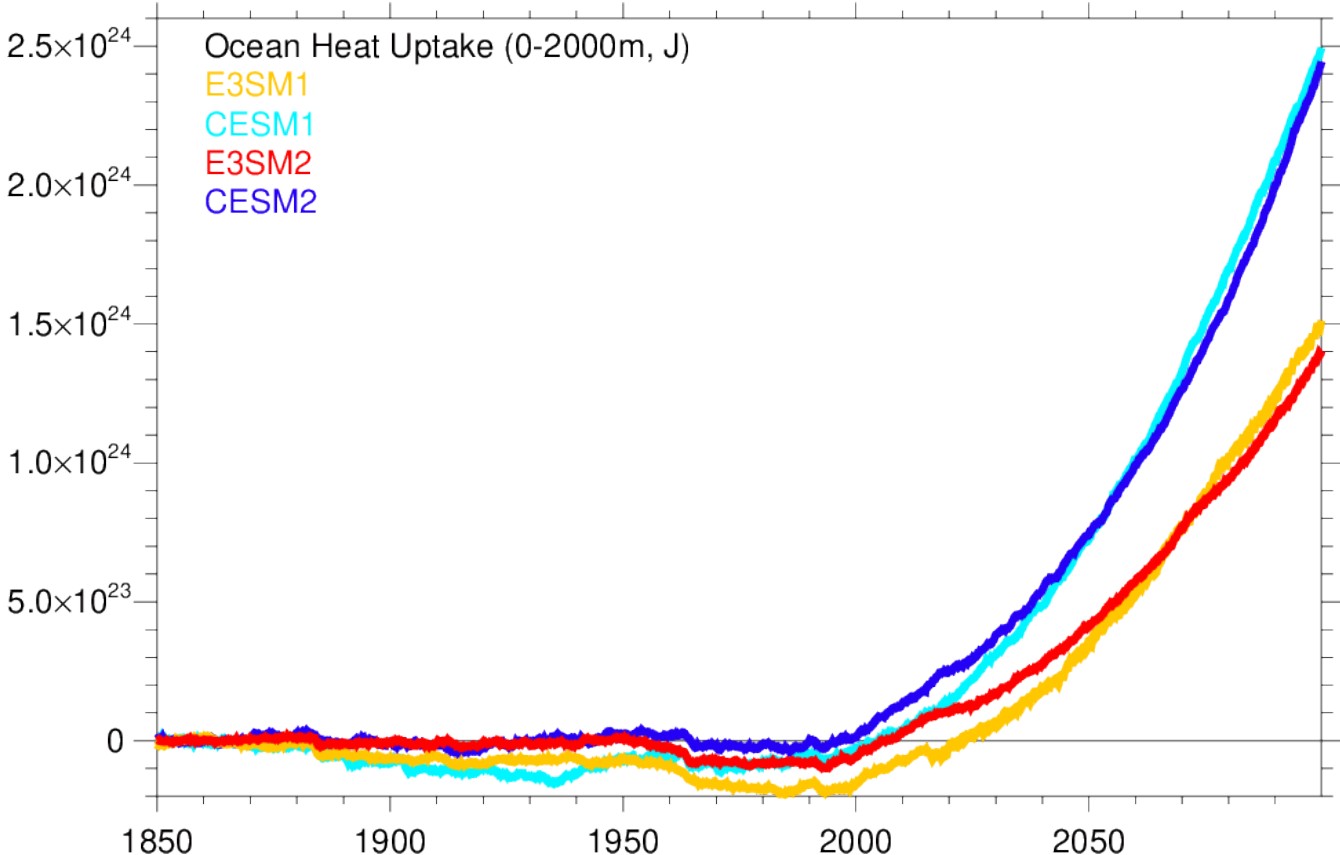

**Figure 11: Ensemble-mean zonal mean ocean heat content change (J) from the surface to 2000 m versus the 1850-59 average in E3SM1 (a) CESM1 (b) E3SM2 (c), and CESM2 (d). Drifts have been removed from each time series.**



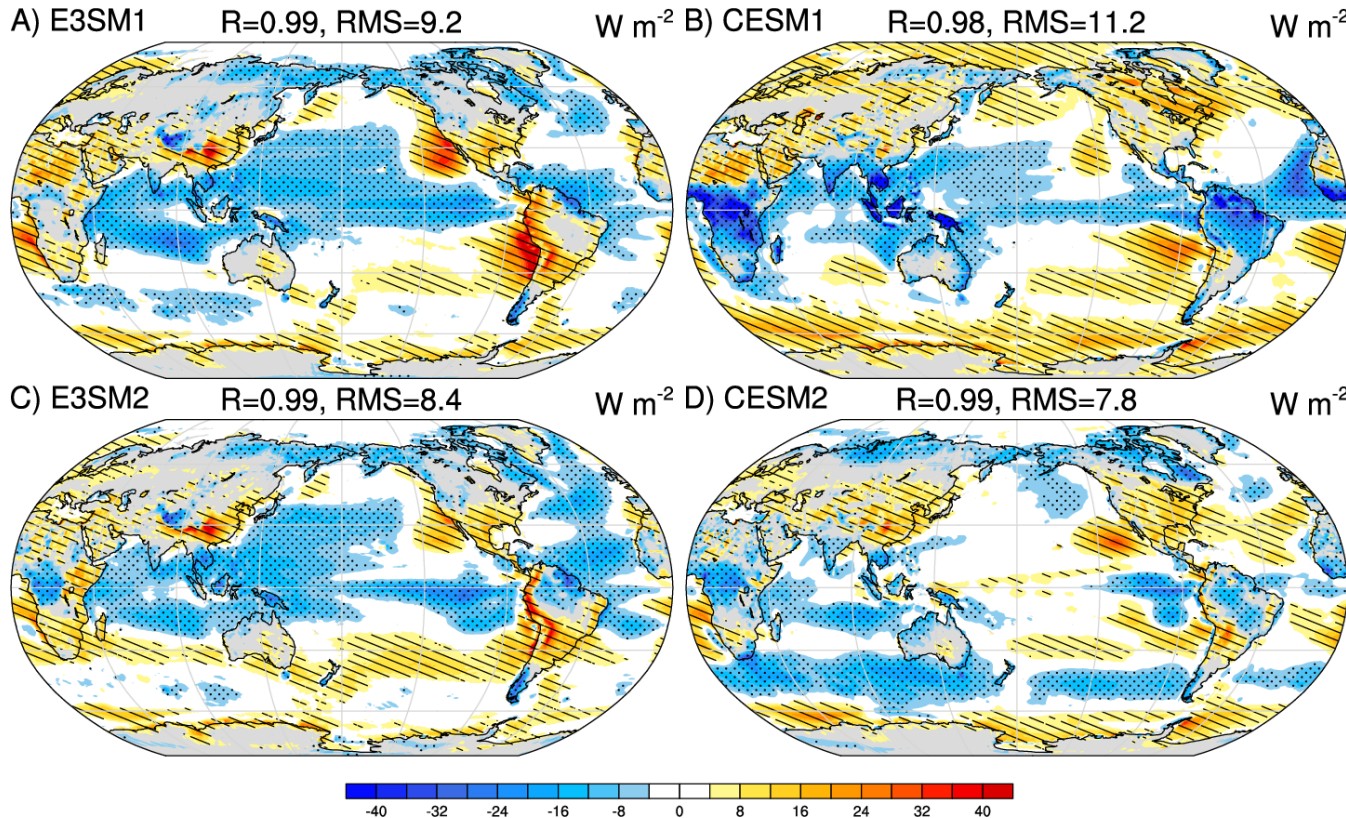

**Figure 12: Climatological ensemble-mean (2000-2020) net top-of-model radiation (R$_T$) biases relative to CERES estimates from E3SM1 (a), CESM1 (b), E3SM2 (c), and CESM2 (d). Pattern correlation (r) and root-mean-squared error (RMSE) between the models and CERES are also shown in the title for each panel. Hatching and stippling corresponds to biases greater than 10 W m$^{-2}$, and less than -10, W m$^{-2}$, respectively.**



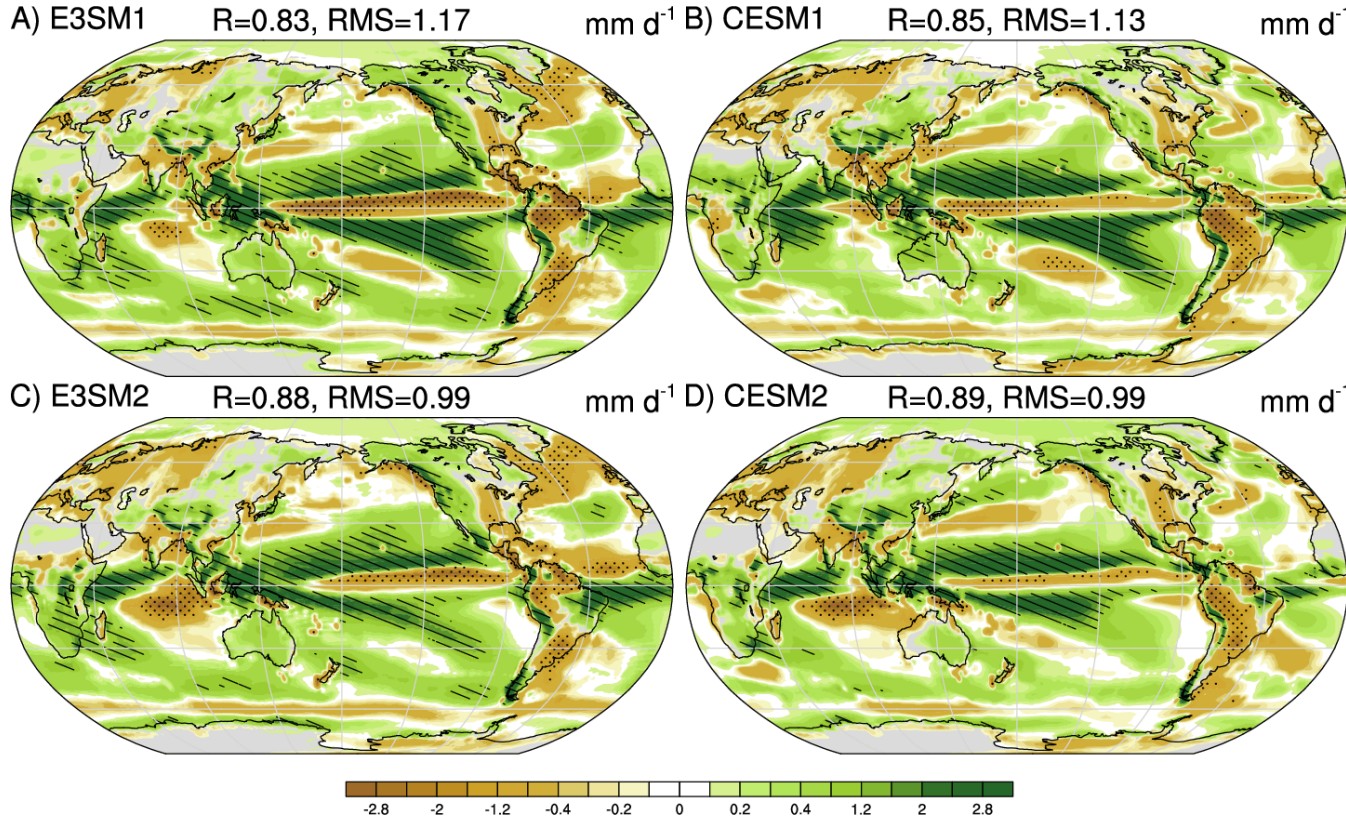

**Figure 13: Climatological ensemble-mean (1979-2020) precipitation biases relative to GPCP estimates from E3SM1 (a), CESM1 (b), E3SM2 (c), and CESM2 (d). Pattern correlation (r) and root-mean-squared error (RMSE) between the models and GPCP are also indicated in the title for each panel. Hatching and stippling corresponds to biases greater than 1 mm day⁻¹, and less than -1 mm day⁻¹, respectively.**






**Figure 14: Monthly (bars) and 12-mo running mean (solid line) ensemble-mean responses to variable biomass emissions in E3SM2 for (a) cloudy-sky albedo, (b) T$_{2m}$, (c) SW$_{TOM}$ (c), and SWSFC (d). The associated sensitivities of CESM2 (12-mo running mean) are also shown (dashed lines). (e) The spatial pattern of warming in response to CMIP6 biomass emissions (versus smoothed).**



**Code availability**

The CMATv1 code has been made available at doi: 10.5194/gmd-13-3627-2020.

**Data availability**

The data for this study will be made available on the Global Data Exchange at http://gdex.ucar.edu. The CERES data can be accessed at https://ceres.larc.nasa.gov/data/

**Author contributions**

JF and JC designed the study. JF and JC carried out the analysis and drafted the first version of the manuscript. All authors
contributed to structuring the analysis and reviewing the manuscript.

**Competing interests**

The authors have no competing interests to declare.

**Disclaimer**

Publisher's note: Copernicus Publications remains neutral with regard to jurisdictional claims in published maps and institutional affiliations.

**Acknowledgements**
TBD.

**Financial support**

The efforts of Dr. Fasullo in this work were supported by NASA Awards 80NSSC21K1191, 80NSSC17K0565, and 80NSSC22K0046, and by the Regional and Global Model Analysis (RGMA) component of the Earth and Environmental
System Modeling Program of the U.S. Department of Energy's Office of Biological & Environmental Research (BER) under Award Number DE-SC0022070. This work also was supported by the National Center for Atmospheric Research, which is a major facility sponsored by the National Science Foundation (NSF) under Cooperative Agreement No. 1852977. Dr. Fasullo was also supported by NSF Award 2103843.
