# Peer review of "An Overview of the E3SM version 2 Large Ensemble and Comparison to other E3SM and CESM Large Ensembles"

_EGUsphere, 2023_

## Author Response (AR1)

**National Center for Atmospheric Research**
**Climate and Global Dynamics Division**
**Climate Analysis Section**

**NCAR**

***Dr. John T. Fasullo***
*fasullo@ucar.edu, http://www.cgd.ucar.edu/staff/fasullo/index.html*
*P. O. Box 3000 • Boulder, CO  80301*
*Tel: 303-497-1712 • Fax: 303-497-1333*

26 Jan 2024

To: *Editor, Earth System Dynamics*

From: Dr. John Fasullo

**Subject: Submission of a Research Article**

Attached please find our submission entitled "**An Overview of the E3SM version 2 Large Ensemble and Comparison to other E3SM and CESM Large Ensembles**", with minor revisions. We appreciate your efforts and those of two anonymous reviewers during the review process. Thanks!

Sincerely,

John Fasullo

**Editor Comments:**

It would be very useful to provide more analysis on why the model is unable to correctly reproduce basic features of the t2m evolution in historic period and how it is (or it is not) related to an excessive reaction to the aerosol forcing. Is it possible that the ocean is also involved in the process?

**Details of this effect are covered in Golaz et al. 2022 and arise from the excessive brightening of clouds by aerosols. That said the editor raises the valid point that we have the opportunity to better clarify this and to illustrate connections to the model's weak AMOC, which reduces the model's ability to buffer the surface temperature response to these and other radiative forcing anomalies. We have modified the manuscript to make these points.**